# Novel Nanoprobe with Combined Ultrasonography/Chemical Exchange Saturation Transfer Magnetic Resonance Imaging for Precise Diagnosis of Tumors

**DOI:** 10.3390/pharmaceutics15122693

**Published:** 2023-11-28

**Authors:** Jieqiong Ding, Liu He, Lin Yang, Liyuan Cheng, Zhiwei Zhao, Binhua Luo, Yanlong Jia

**Affiliations:** 1Xianning Medical College, Hubei University of Science and Technology, Xianning 437100, China; djq197911@163.com (J.D.); bjxjlzxhl@163.com (L.H.); 17861407689@163.com (L.C.); 2Department of Radiology, Second Affiliated Hospital, Shantou University Medical College, Shantou 515041, China; yanglin2019fuer@163.com; 3Department of Radiology, Xianning Central Hospital, The First Affiliated Hospital of Hubei University of Science and Technology, Xianning 437100, China; fly123sky@163.com; 4Department of Radiology, Xiangyang Central Hospital, Affiliated Hospital of Hubei University of Arts and Science, Xiangyang 441021, China

**Keywords:** PEG-PLLA, nanoprobe, nanoparticle, US imaging, CEST MR imaging

## Abstract

Given that cancer mortality is usually due to a late diagnosis, early detection is crucial to improve the patient’s results and prevent cancer-related death. Imaging technology based on novel nanomaterials has attracted much attention for early-stage cancer diagnosis. In this study, a new block copolymer, poly(ethylene glycol)-poly(l-lactide) diblock copolymer (PEG-PLLA), was synthesized by the ring-opening polymerization method and thoroughly characterized using Fourier transform infrared spectroscopy (FT-IR), proton nuclear magnetic resonance spectroscopy (H-NMR), X-ray diffraction (XRD), and thermogravimetric analysis (TGA). The obtained PEG-PLLA was used to prepare nanoparticles encapsulated with perfluoropentane and salicylic acid by the emulsion-solvent evaporation method, resulting in a new dual-mode nano-image probe (PEG-PLLA@SA·PFP). The zeta potential and mean diameter of the obtained nanoparticles were measured using dynamic light scattering (DLS) with a Malvern Zetersizer Nano. The in vitro biocompatibility of the PEG-PLLA nanoparticles was evaluated with cell migration, hemolysis, and cytotoxicity assays. Ultrasonic imaging was performed using an ultrasonic imaging apparatus, and chemical exchange saturation transfer (CEST) MRI was conducted on a 7.0 T animal scanner. The results of IR and NMR confirmed that the PEG-PLLA was successfully synthesized. The particle size and negative charge of the nanoparticles were 223.8 ± 2.5 nm and −39.6 ± 1.9 mV, respectively. The polydispersity of the diameter was 0.153 ± 0.020. These nanoparticles possessed good stability at 4 °C for about one month. The results of cytotoxicity, cell migration, and hemolysis assays showed that the carrier material was biocompatible. Finally, PEG-PLLA nanoparticles were able to significantly enhance the imaging effect of tumors by the irradiation of ultrasound and saturation by a radiofrequency pulse, respectively. In conclusion, these nanoparticles exhibit promising dual-mode capabilities for US/CEST MR imaging.

## 1. Introduction

Imaging techniques play a pivotal role in clinical practice, as they can provide crucial information for the early pathological diagnosis of diseases, particularly tumors, which can increase the chances of successful treatment and significantly prolong patient survival [1]. However, due to the diversity of tumor pathology and the complexity of the anatomical structures, imaging after surgical resection and radiotherapy can be a significant challenge, which makes it difficult for a single imaging model to perform hierarchical diagnosis and post-treatment monitoring of tumors. To address these challenges, multimodal imaging technologies have been developed and applied in the clinic to provide more comprehensive and specific information for accurate cancer diagnosis and treatment monitoring.

The acquisition processes for some of the most popular non-invasive imaging instruments, including computed tomography (CT) and positron emission tomography (PET), require the use of ionizing radiation, whereas the procedures for the other two techniques, ultrasound (US) molecular imaging and MRI, are non-ionizing and regarded as safe. By combining different imaging modalities, multimodal imaging technologies can achieve advantages that are difficult to achieve with a single imaging modality. For example, MRI–PET dual-modal imaging can achieve both metabolic and morphological tumor characterization using complementary PET and MRI data. CT–US can be used to image both hard and soft tissues with high resolution in vivo and in vitro, providing comprehensive morphological information for tumor diagnosis. In addition, MRI–US dual-modal imaging can combine the high soft tissue resolution of MRI with real-time US images, thereby achieving highly accurate diagnoses of cancer and other soft tissue diseases. They are mutually complementary, which provides a better understanding of the anatomy or an improved evaluation of the therapeutic outcomes.

US molecular imaging, a type of non-invasive molecular imaging technology, has the potential to image tumor tissues more accurately and sensitively [2,3,4]. US molecular imaging can detect tumor tissues by using the characteristics of ultrasound scattering and absorption of tissues. It can also use contrast agents to enhance the ultrasound signal, improve the detection sensitivity, and reduce the interference of background noise. Perfluoropentane is a perfluorinated organic compound that is a colorless and transparent liquid at room temperature. It has characteristics such as a low boiling point, high dielectric constant, and easy preparation of uniform small bubbles. It is often used as a contrast agent for ultrasound contrast in clinical practice. Contrast-enhanced ultrasonic imaging, which may significantly boost the picture contrast and spatial resolution, uses contrast agents, such as microbubbles encasing gases with a diameter of around 1–8 μm [5]. Although they may significantly enhance imaging contrast and quality, their lack of stability prevents them from being used in clinical settings. Moreover, the microbubbles can only be utilized as blood pool contrast agents since they cannot enter tumor arteries or tumor tissues to accomplish tissue imaging [6,7]. Therefore, US molecular imaging has great potential in the field of molecular imaging, but it needs to be combined with other imaging technologies to achieve better application and development.

Magnetic resonance imaging (MRI) is a powerful, non-invasive medical imaging technique that can provide anatomical details without the need for ionizing radiation or harmful radionuclides for soft tissue imaging [8,9]. The most routinely used compounds for contrast enhancement are gadolinium-based contrast agents, which can significantly shorten the T1 relaxation time of the targeted tissues and display a brightened region. However, these agents have several disadvantages, such as a short imaging time, fast diffusion, and the risk of nephrogenic systemic fibrosis [10,11,12]. Iron oxide nanoparticles are another kind of widely adopted magnetic nanoprobes used as T2/T2* MRI contrast agents, which have a high relaxation time. However, iron oxide nanoparticles in the blood circulation system are generally hydrophobic and susceptible to phagocytosis and are cleared by the reticuloendothelial system (RES) [13].

Over the past decade, Chemical Exchange Saturation Transfer Technology (CEST) MRI techniques have attracted interest as a robust imaging approach for probing molecular imaging changes in vivo [14,15,16,17,18]. CEST is a new MRI approach based on the theory of magnetization transfer (MT) and chemical exchange. It utilizes a selective radiofrequency (RF) irradiation pulse on exchangeable protons (such as -OH, -NH, and -NH_2_), resulting in a loss of the bulk water proton signal intensity [19,20,21,22]. By detecting the degree of reduction of free water signals, CEST can obtain significant physiological and biochemical information about the target tissue. CEST not only enables pH imaging but is also a type of non-invasive imaging suitable for studying tumor metabolism and the tumor microenvironment [23,24,25,26,27,28]. To further enhance the imaging effect, contrast agents are often employed [29,30,31]. Contrast media can be divided into endogenous and exogenous. Endogenous contrast agents in the body, such as polypeptides and amino acids, can only exchange a few hydrogen molecules with water, resulting in a relatively low imaging effect [32]. Therefore, it is very important to develop an artificial substance, an exogenous contrast agent that can exchange more protons and improve the imaging effect [33].

Salicylic acid, which is used as a contrast agent in CEST imaging, can produce better imaging because the resonant frequency of hydration of its carboxyl group and phenolic hydroxyl group differ from one another, and the latter is higher than water and can exchange considerable amounts of protons with water [34,35]. However, the accumulation of salicylic acid in vivo may pose harm to the ears, necessitating the reduction of its side effects [36,37,38].

Polylactic acid (PLLA) has been verified as a non-toxic, non-carcinogenic, and non-teratogenic biodegradable polymer. Its metabolites are carbon dioxide and water, which circulate in the body as lactic acid [39,40,41,42]. PLLA is also widely used in microcapsules, microspheres, nanoparticles, and other drug delivery systems. Synthesizing PLLA mainly involves direct polymerization and ring-opening reactions. The molecular weight of PLLA acquired by the latter is more controllable than the former [43]. However, this polymer displays hydrophilicity, making it challenging to disperse evenly in water [44]. Polyethylene glycol (PEG) is a hydrophilic, nontoxic polymer material with good stability in vivo and it is widely used in injections, tablets, eye drops, etc. PEG with a high molecular weight is not metabolized in vivo and is directly excreted through urine. The –OH of PEG can provide a polymerization reaction with the active site, and the reaction product is a block copolymer. If PEG-PLLA amphiphilic block copolymers can be prepared, PLLA will not only retain its biocompatibility but will also improve its hydrophilicity.

In this study, a new amphiphilic block copolymer, PEG-PLLA, was prepared by ring-opening polymerization, in which PEG opened the six-membered ring of L-lactide under the catalysis of stannous octoate dissolved in dichloromethane. The nanoparticles loaded with salicylic acid and perfluoropentane were prepared using the emulsion and solvent evaporation method, as shown in Figure 1. The physicochemical characteristics of the nanoparticles, including their appearance, particle size, and suspension stability, were assessed. Additionally, the nanoparticles were evaluated for their performance in vitro and in vivo US/CEST MR dual-mode imaging.

## 2. Materials and Methods

### 2.1. Materials and Reagents

L-lactide (optical purity of 95%) was obtained from Daigang Biomaterial Co., Ltd. (Jinan, China). Polyethylene glycol (average Mn = 6000) was purchased from Aladdin Chemistry (Shanghai, China). Stannous octoate was purchased from Sigma Chemical Co. (St. Louis, MO, USA). Perfluoropentane (PFP) was purchased from JenKem Technology Co., Ltd. (Beijing, China). Trichloromethane, diethyl ether, salicylic acid, absolute ethyl alcohol, Tween 80, Span 80, and lauryl sodium sulfate (SDS) were purchased from Sinopharm Group Chemical Reagent Co., Ltd. (Shanghai, China).

### 2.2. The Synthesis of PEG-PLLA Diblock Copolymer

A certain amount of L-lactide (recrystallized three times with ethyl acetate), PEG6000 and stannous octanoate (the molar ratio was 100:1:1) were added into a 50 mL round-bottom flask. Then, the mixture was reacted for 2 h in an oil bath at 120 °C under the protection of dry high-purity nitrogen. The obtained product (PEG-PLLA) was dissolved in chloroform and precipitated in excess cold diethyl ether. Finally, the sediment was separated and dried for 24 h under vacuum conditions at 40 °C. The procedure was repeated three times. The route of synthesis is shown in Figure 2.

### 2.3. Structural Characterization of PEG-PLLA Diblock Copolymer

The FT-IR spectra of PEG-PLLA, PEG, and PLLA were performed with an infrared spectrophotometer (IRAFFINITY-1, Shimadzu, Kyoto, Japan). The samples were respectively dried with an infrared lamp and mixed with dried KBr. The mixed powders were tableted and measured from wavelength 4000 cm^−1^ to 400 cm^−1^.

The ^1^H-nuclear magnetic resonance (^1^H-NMR) spectra of samples were measured with a Bruker Avance III HD 400 MHz spectrometer. The PEG-PLLA copolymer was dissolved in CDCl_3_ and tetramethylsilane (TMS) was used as the standard substance.

The X-ray diffraction (XRD) patterns of the polymers (PLLA, PEG, and PEG-PLLA) were measured with an X-ray diffractometer (XRD-6100, Shimadzu, Japan). The current of the tube was set at 30.0 mA with a tube voltage of 40.0 kV, and the scan range of 2θ was 10°~80°.

The thermogravimetric (TG) curves of the samples (PLLA, PEG, and PEG-PLLA) were measured by thermogravimetry (TG209F3, Netzsch, Hanau, Germany). The temperature range was set at 100~600 °C under nitrogen flow with a heating rate of 10 °C/min.

### 2.4. Preparation of PEG-PLLA Nanoparticles Encapsulating Salicylic Acid and Perfluoropentane

PEG-PLLA (50.0 mg) was dissolved in 1 mL of a chloroform solution containing two drops of Span 80. Then, 30.0 mg of salicylic acid was dissolved in 0.5 mL absolute ethyl alcohol, which was mixed with the copolymer solution, and 500 μL of PFP was added to the mixed solution in an ice bath. Subsequently, the hybrid solution was stirred vigorously at 12,000 rpm for 5 min to form a coarse emulsion. Then, the obtained emulsion was added drop by drop to a 0.5% sodium dodecyl sulfate solution (5 mL) under the action of ultrasound at 195 W in an ice bath, and sonication lasted for 5 min under these conditions. Finally, the resulting liquid was stirred for 24 h at room temperature. The PEG-PLLA nanoparticles encapsulating salicylic acid and PFP were centrifuged and dispersed in a normal saline solution.

### 2.5. Characterization of PEG-PLLA Nanoparticles Encapsulating Salicylic Acid and Perfluoropentane

The mean particle size and zeta potential of the nanoparticles were determined by dynamic light scattering (Malvern, Malvern, UK), and the samples were diluted with distilled water and measured three times at 25 °C. The morphology of the nanoparticles was examined by transmission electron microscopy (TEM).

### 2.6. Cell Toxicity Assay

The cytotoxicity of the PEG-PLLA copolymer was determined by Cell Counting Kit-8 (CCK-8) (Wuhan Kerui Biotechnology Co., Ltd, Wuhan, China) assays with rat glioma cells (C6 cells, purchased from the Shanghai Institute of Life Science Cell Culture Center, Shanghai, China). Firstly, 1 mL of DMEM culture medium containing 10% FBS was used for cell resuscitation in an atmosphere of 5% CO_2_ at 37 °C. The cells were passaged 3 times before use in the toxicity assay. The C6 cells were distributed uniformly into a 96-well plate with 6000 cells/well. After culture overnight, the cell medium was replaced with 200 μL medium containing different concentrations of PEG-PLLA copolymer. After 24 h of incubation, 10 μL of CCK-8 solution was added to each well and further incubated for 1 h. Finally, the absorbance values were detected at a wavelength of 492 nm with a microplate reader.

### 2.7. Cell Migration Test

The cell migration rate was measured by a cell scratch test. When the density of the C6 cells was about 90%, the tip of a 200 μL sterile pipetting gun was held vertically to the six-well plate and used to scrape lines along the cell monolayer. The obtained residues were washed with PBS, and the cells were supplemented with fresh culture medium. Then, the cells were cultured in the incubator for 20 min and photographed by an inverted microscope (Olympus, Tokyo, Japan), which was used as an image of 0 h. Subsequently, different concentrations of PEG-PLLA (0.25, 0.5, 0.75, and 1.00 μg/μL) were added to the plates, and a control group that did not contain any material was set at the same time. Finally, the cells were cultured for another 24 h and photographed. The migration rate of the C6 cells was calculated using Image J software (https://imagej.nih.gov/ij/download.html (accessed on 26 September 2023)), and the results were analyzed using Graph Pad Prism software 8.0.0 (San Diego, CA, USA).

### 2.8. Hemolysis

The hemolysis assay in vitro was performed with arterial blood, which was taken from the thigh artery of male rats and centrifuged at 4 °C and 1500 rpm for 5 min to collect the blood cells. The cell pellet was washed with normal saline several times until the liquid supernatant was clear and then the cells were redispersed with normal saline to obtain a 2% (*v*/*v*) suspension. Nanoparticle suspensions were added to a 2% (*v*/*v*) erythrocytes suspension to produce mixtures at various concentrations (0.05, 0.25, 0.5, 0.75, and 1.00 μg/mL). Double-distilled water as a positive control sample and normal saline as a negative control sample were used with the same process as the nanoparticles. Then, the samples were cultured at 37 °C for 1 h followed by centrifugation at 1000 rpm and 4 °C for 5 min. At last, the liquid supernatants were collected and distributed evenly in 96-well plates, and the absorbance was measured at 540 nm with a spectrophotometer. The percentage of hemolysis was obtained by using the following equation:$$\mathrm{Hemolysis}\;\mathrm{ratio}=\frac{A_{\mathrm{sample}}-A_{\mathrm{negative}}}{\;A_{\mathrm{positive}}-A_{\mathrm{negative}}}\times 100\%$$

### 2.9. Ultrasound Imaging In Vitro

The ultrasound contrast imaging in vitro was conducted with an ultrasonic imaging apparatus (Philips IU22, Amsterdam, The Netherlands) with a 10 MHz probe. The PEG-PLLA nanoparticles were placed in the finger of a latex rubber glove, which was immersed in a water bath. A 10 MHz probe approached the finger of the glove and achieved imaging by the ultrasonic equipment when the temperature of the water bath was 20, 22, 24, 26, 28, 30, 32, 34, 36, and 38 °C, respectively. Normal saline and the coupling agent served as blank control groups.

### 2.10. Animal Care

All animal care and experiments were performed according to the Guidelines of Animal Experimentation of Hubei University of Science and Technology and the National Institute of Health. The experiment was approved by the Institutional Animal Care and Use Committee of Hubei University of Science and Technology (approval number: 2022-0023).

### 2.11. Ultrasound Imaging In Vivo

Male BALB/c nude mice (about 20 g) were used to construct the animal models of a subcutaneous tumor. Suspensions of C6 cells (about 3 × 10^6^ cells per mouse) were subcutaneously inoculated into the back of the nude mice. When the volume of the tumor reached approximately 1000 mm^3^, ultrasound images were recorded as a control with a 10 MHz probe and the mechanical index was set at 0.08. Then, the nanoparticle solution (150 µL) was injected into the tumor, and contrast-enhanced ultrasound images were collected with the same parameters.

### 2.12. CEST MR Imaging In Vitro

This experiment was conducted with a 7.0 T animal magnetic resonance scanner (Agilent, Santa Clara, CA, USA). The PEG-PLLA nanoparticles solution was diluted with PBS and deionized water to prepare solutions (1 mg/mL) of pH = 6.0, 6.4, 7.0, and 7.6 at ambient temperature. Then, they were placed in 1 mL test tubes. In addition, the test tubes were fastened with 1% agarose prepared with deionized water, which could eliminate artifacts caused by air near the test tubes. All of the samples were measured by irradiation with a saturated radio frequency (RF) pulse whose intensity was respectively 1.0, 2.0, 2.5, 3.0, and 4.0 μT. Scan parameters of the EPI-CEST were set as follows: TR = 4000 ms, TE = 10 ms, FOV = 30 mm × 30 mm, matrix size: 128 × 128, average: 4, segments/ETL = 16/8, Kzero = 4.

### 2.13. CEST MR Imaging In Vivo

SD rats (n = 10) were used to construct a glioma model using a method described in our previous report 11. Firstly, the rats were anesthetized with an injection of 10% chloral hydrate (0.4 mL/100 g) and fixed in a stereotactic frame in a prone position. The right frontal lobe was selected as the site of glioma cell inoculation by cutting along the middle line of the head of the rat with eye scissors. A microsyringe containing 10 μL C6 cell suspension was fixed on a three-dimensional scaffold after the right frontal cavity was drilled. A cell suspension (2 μL) was gently injected into the inoculation site and left for 2 min to observe the status of the rats. We repeated these steps until the 10 μL C6 cell suspension was completely injected. Then, after about 5 min, the microsyringe was slowly drawn out. Finally, the wound was disinfected and sutured. After the glioma grew in the brain of rats for about 2 weeks, CEST imaging could be performed.

Glioma rats were anesthetized with 10% chloral hydrate (0.4 mL/100 g) and fixed in the standard coil with a prone position. Then, EPI-CEST imaging technology was used to scan the brains of the rats. At 30, 45, 75, 90, 105, 120, 135, and 150 min after PEG-PLLA nanoparticles containing perfluoropentane and salicylic acid (150 μL) were injected into the caudal vein, CEST MR imaging was performed. A high-resolution T2-weighted axial slice crossing the center of the tumors was acquired with TR = 4000 ms, TE = 10 ms, slice thickness = 2 mm, FOV = 30 mm × 30 mm, matrix size = 128 × 128, segments/ETL = 16/8, Kzero = 4. For in vivo CEST imaging, the parameters were as follows: TR = 6000 ms, TE = 27.63 ms, slice thickness = 3 mm, FOV = 40 mm× 40 mm, matrix size = 64 × 64, ETL = 64, dummy scans = 7, saturation power = 3.6 μT, saturation time = 3 s, with 122 frequency offsets unevenly distributed from −15 to 15 ppm relative to the resonance of water. The total scanning duration was 13 min.

All CEST image processing and data analysis were performed using custom-written scripts in MATLAB (Mathworks, Natick, MA, USA, R2016b). The Lorentz fit method was applied to obtain CEST images. The multi-pool Lorenz model equation is as follows:$$L\left(A_{1},{\;L}_{W},\;∆\omega \right)=\frac{A*{L_{W}}^{2}/4}{\frac{{L_{W}}^{2}}{4}+∆\omega^{2}}=\frac{A}{1+4{\left(\frac{∆\omega }{L_{W}}\right)}^{2}}$$
where L_W_ is the bandwidth, A is the amplitude, and Δω is the relative displacement frequency with respect to the center frequency of the curve at which the CEST effect is the amplitude at a chemical shift of 9.3 ppm.

### 2.14. Statistical Verification

Statistical verification was carried out with GraphPad Prism 6.0. When the *p* value was less than 0.05, this indicated that there was a significant difference.

## 3. Results and Discussion

### 3.1. Characteristics of PEG-PLLA Copolymer

The infrared spectra of PEG, PLLA, and PEG-PLLA are shown in Figure 3. Compared with the IR spectra of PEG, a new strong absorption peak corresponding to the carbonyl group stretching vibrations was observed in the IR spectra of the PEG-PLLA copolymer at approximately 1765 cm^−1^; weak stretching bands of -CH_3_, -CH_2_, and -CH were observed at 2988 cm^−1^, 2875 cm^−1^, and 2926 cm^−1^, respectively; and the band at 1100 cm^−1^ was attributed to asymmetrical stretching of the C-O-C ester, which provided evidence of the formation of the ester group. The peaks at 1465 cm^−1^ and 1375 cm^−1^, respectively, corresponded to methylene and methyl, and the wavenumber at 1050 cm^−1^ corresponded to the stretching vibration band of C-O. Based on the above results, it could be concluded that the copolymer of PEG-PLLA was successfully synthesized.

The ^1^H-NMR spectrum of PEG-PLLA is shown in Figure 4. The peak at about 1.5 ppm was a chemical shift (δ) of the -CH_3_ group in the PLLA main chains, which presented as a double peak. The chemical shift of the -CH group in the PLLA main chains was 5.2 ppm due to the effect of the adjacent carbonyl group. The characteristic peak of –CH_2_ in the PEG chains was observed at 3.6 ppm, which was affected by the electron-absorbing effect of the O atom. These results confirmed the successful preparation of PEG-PLLA. In addition, the peak area in the H-NMR spectrum was proportional to the number of H, and the molecular weight of PEG was 6000. The molecular weight and degree of polymerization of PLLA could be calculated based on the ^1^H-NMR spectrum of PEG-PLLA. Finally, the polymerization degree of PLLA was estimated to be 207, and its molecular weight was about 14,904.

The XRD patterns of PLLA, PEG, and PEG-PLLA are shown in Figure 5. The diffraction peaks of PLLA mainly appeared at 16.7°, 19.2°, and 22.5°, indicating that the PLLA mainly formed α crystals. The 2θs of PEG were mainly 19.2° and 23.4°, which suggested that the PEG polymer displayed a monoclinic crystal. Compared with PLLA, the diffraction peaks at 16.7°, 19.2°, and 22.5° for the PEG-PLLA copolymer changed slightly; however, the diffraction peaks at 44.2°, 64.3°, and 77.4°disappeared entirely, indicating that the addition of PEG destroyed the crystal structure of PLLA, causing a decrease in the orderliness of the PLLA molecules.

The TG curves of PLLA, PEG, and PEG-PLLA are shown in Figure 6. As can be seen from the figure, the mass of PLLA began to decrease at 137 °C, while the same phenomenon occurred at 162 °C for the copolymer, which indicated that the thermal stability of the copolymer was better than that of PLLA, and the addition of PEG enhanced the thermal stability of PLLA. In addition, there were two plateaus in the TG curves of PEG-PLLA, and the boundary between the two plateaus was 34%. However, the mass ratio of L-lactide and polyethylene glycol in the raw material formula of the copolymer was 2:1, which was consistent with the thermogravimetric curve.

### 3.2. Measurement of PEG-PLLA@SA·PFP Nanoparticle Encapsulating Salicylic Acid/PFP Characteristics

The nanoparticle suspension was uniform and milk-white, as shown in Figure 7a. The average diameter of the PEG-PLLA@SA·PFP nanoparticles was measured by a Malvern laser particle size instrument, as shown in Figure 7b. The average size of the nanoparticles was 223.8 ± 2.5 nm, and the PDI (polydispersity index) was 0.153 ± 0.020, indicating that the particle size of the nanoparticles was relatively uniform. The zeta potential of the nanoparticles was a negative charge of −39.6 ± 1.9 mV. The PEG-PLLA@SA·PFP nanoparticles were stored in the refrigerator at 4 °C for 30 days to assess their stability by measuring their diameter and potential with dynamic light scattering. As shown in Figure 7c, there was no significant variation in the diameter or zeta potential over the course of 30 days (0, 5, 10, 15, 20, 25, and 30 days), which confirmed that the PEG-PLLA@SA·PFP nanoparticles were relatively stable. TEM images of the PEG-PLLA@SA·PFP nanoparticles are shown in Figure 7d.

### 3.3. Cytotoxicity Analysis

The cytotoxicity of the PEG-PLLA copolymer was determined by a CCK-8 assay. C6 cells were cultured with the copolymer of PEG-PLLA under different concentrations for 24 h, and the results are shown in Figure 8. These results indicated that the copolymer of PEG-PLLA prepared in this study had little effect on the viability of C6 cells compared to that of the blank control.

### 3.4. Cell Migration Analysis

The effects of the copolymer on the migratory properties of C6 cells were studied using scratch assays. At four concentrations of PEG-PLLA (0.25, 0.5, 0.75, and 1.00 μg/μL), there was no significant difference in cell migration, indicating that PEG-PLLA as an extracellular matrix did not affect cell migration. These results are displayed in Figure 9. The cell migration rate was calculated by the following equation:$$\mathrm{Migration}\;\mathrm{rate}=\left[\left(A_{0}-A_{t}\right)/A_{0}\right]\times 100\%$$

The cellular migration ratio of the control group (without PEG-PLLA) was 76.15 ± 1.16%, and the sample groups of test concentrations (0.25, 0.5, 0.75, and 1.00 μg/μL) were respectively 76.10 ± 1.35%, 75.64 ± 1.22%, 75.29 ± 1.62%, and 74.61 ± 1.25% (n = 4). Statistical testing found the results showed no significant differences (*p* > 0.05), as shown in Figure 10.

### 3.5. Hemolysis Assay of PEG-PLLA

Hemolysis is a dangerous condition where blood cells rupture in circulation, which can eventually lead to jaundice and anemia. Many natural and synthetic nanoparticles have been found to cause hemolytic action. Therefore, evaluating the hemolytic properties of potential pharmaceutic materials is significant for assessing their biocompatibility. As shown in Figure 11, the hemolysis percentages of all samples at concentrations ranging from 0.05 to 1.0 μg/μL were less than 5%, indicating that the PEG-PLLA is safe for erythrocytes and could be used as a pharmaceutic adjuvant.

### 3.6. Ultrasound and Heat-Triggered Vaporization of Perfluoropentane

Under continual ultrasound exposure, nanoparticles containing gas are constantly compressed and swelled by positive and negative pressure, respectively, and they produce signals of backscattering. The intensity of the acoustic backscatter signal is proportional to the square of the difference between the compressibility of the nanoparticles and the medium. The compression ratio of gas is several orders of magnitude larger than that of blood or soft tissue, which makes the high acoustic impedance mismatch between them and nanoparticles useful as contrast agents for ultrasound imaging. Perfluoropentane (PFP) is in a liquid state at room temperature and atmospheric pressure, and its boiling temperature is 28.5 °C. However, when PFP is encapsulated by carrier materials, its boiling point will increase greatly as a result of Laplace pressure.

According to the literature, the two main factors affecting PFP gasification are temperature and ultrasonic waves [6,45]. However, in our experiment, we found that when the temperature increased to 40 °C, the particle size of the nanoparticles hardly changed, indicating that the wrapped PFP did not undergo a phase transition, and vaporization of PFP via heating was inefficient and required temperatures that were well above the physiological range. In what follows, we explored the synergistic effect of ultrasound and temperature as triggering factors for the phase transition of PFP.

The nanoparticle solution was placed in a sealed finger of a latex glove, which was immersed in a thermostatic water bath. The initial temperature of the water bath was set at 20 °C, and after two minutes at constant temperature, the ultrasonic probe touched the latex gloves to collect an image. Then, the temperature was raised to 22 °C and maintained for two minutes, and an image was taken. Similarly, the temperature was gradually increased to 38 °C, and images were collected separately. As shown in Figure 12 and Figure 13, there was a dark area in the image managed at 20 °C, and the contrast was lower. However, as the temperature gradually increased, more bright spots appeared in the images, and the difference was significantly improved. However, when the temperature was 38 °C, the intensity of the signal began to decrease. These results demonstrated that the nanoparticles were well-suited for imaging at 37 °C.

### 3.7. Echogenicity of Nanoparticles In Vitro at 37 °C

A sealed finger of the latex glove containing about 3 mL of the nanoparticle solution was immersed in water at 37 °C. An ultrasonic imaging system was used to record contrast-enhanced images on B-mode with a 10 MHz probe. Then, saline and the coupling agent were respectively poured into the fingers of the gloves and images were collected under the same experimental conditions. The results are shown in Figure 14. These results showed that the nanoparticles took on an excellent ultrasonic contrast enhancement ability (Figure 14b). However, saline and the coupling agent (Figure 14c,d), which were used as blank control groups, exhibited no ultrasonic contrast enhancement.

### 3.8. Application of Nanoparticles in Ultrasound-Enhanced Imaging of Tumors

To verify the enhanced imaging contrast of PEG-PLLA nanoparticles injected into tumors in vivo, a subcutaneous tumor model of nude mice was successfully prepared. Then, 150 μL nanoparticle solution was injected into the tumors. All contrast images were recorded in the B-mode, with low persistence, a frame rate of 136 Hz, and a mechanical index of 0.4. As shown in Figure 15, there were many dark spots in the ultrasonic images before the injection (Figure 15a), but the contrast of the subcutaneous tumors image was distinctly enhanced after the injection, and the tumors became bright (Figure 15b). This accumulation of nanoparticles can be attributed to their enhanced permeation and retention effect in solid tumors. In addition, the borders between the tumors and normal tissues were more noticeable. These results demonstrated that the nanoparticles could improve the ultrasonic imaging contrast of tumors. These ultrasound images provide a non-invasive method to monitor the accumulation of nanoparticles in subcutaneous tumor tissue in vivo. This technique can be used to assess the targeting ability of different nanoparticles systems and their therapeutic effects on tumors in preclinical studies.

### 3.9. CEST MR Imaging In Vitro

As shown in Figure 16 and Figure 17, the pH and intensity of a saturated RF pulse had a significant effect on the CEST signal strength. According to Figure 16a,b and Figure 17b, it was demonstrated that the intensity of the CEST signal first increased and then decreased as the pH increased. When the pH was about 6.4, the intensity was the strongest. Meanwhile, it was also noted that as the intensity of the RF pulse increased, the CEST signal became stronger, as shown in Figure 16c and Figure 17a. In addition, the signal intensity of water protons irradiated by a saturated RF pulse (S) compared with not irradiated (S_0_) also increased. The S of all samples in the experiment reduced to 0 at 0 ppm. In the meantime, it was shown that the CEST signal at about 9 ppm was caused by the proton chemical exchange of –COOH on the salicylic acid.

### 3.10. CEST MR Imaging In Vivo

Figure 18 displays CEST images in vivo of PEG-PLLA nanoparticles encapsulating perfluoropentane and salicylic acid. Before the intravenous injection of nanoparticles into the mice in this experiment, the brain glioma appeared dark when analyzed by a magnetic resonance scanner, which demonstrated that the signal was lower and the imaging had poor contrast. After the intravenous injection of the nanoparticles, the intensity of the signals began to increase. The strongest signal was at 105 min, and then the intensity began to decrease. This suggests that these nanoparticles have the potential to be used as a contrast agent to enhance the contrast of CEST images for accurate diagnoses of diseases.

## 4. Conclusions

In this study, PEG-PLLA was synthesized by ring-opening polymerization. PEG opened the six-membered ring of L-lactide by catalysis of stannous octoate, and then the PEG-PLLA nanoparticles encapsulating perfluoropentane and salicylic acid were prepared by an emulsification solvent evaporation method. Observation using TEM or DLS revealed a uniform spherical shape and uniform size (223.8 ± 2.5 nm), along with good dispersibility. Multi-scale evaluation using techniques such as FTIR spectroscopy, ^1^H-NMR spectroscopy, XRD, and TGA analysis confirmed the successful synthesis of PEG-PPLA nanoparticles. Furthermore, through cytotoxicity experiments, evaluation of cellular migration, and assessment of hemolysis, we found that these PEG-PLLA nanoparticles displayed no or minimal toxicity and good biocompatibility. Finally, in vitro and in vivo experiments demonstrated that these nanoparticles can serve as a novel contrast agent for ultrasound molecular imaging and CEST MRI, making them promising candidates for dual-mode contrast agent applications. The combination of different imaging modalities in multimodal imaging technologies has broad prospects for clinical applications because it can provide more comprehensive and specific information for accurate cancer diagnosis and treatment monitoring.

## Figures and Tables

**Figure 1 pharmaceutics-15-02693-f001:**
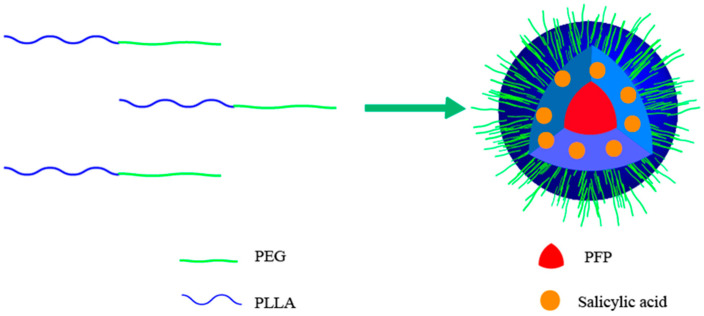
A schematic diagram of a PEG-PLLA@SA·PFP nanoparticle. PEG-PLLA serves as the carrier material to encapsulate SA and PFP to form nanoparticles.

**Figure 2 pharmaceutics-15-02693-f002:**
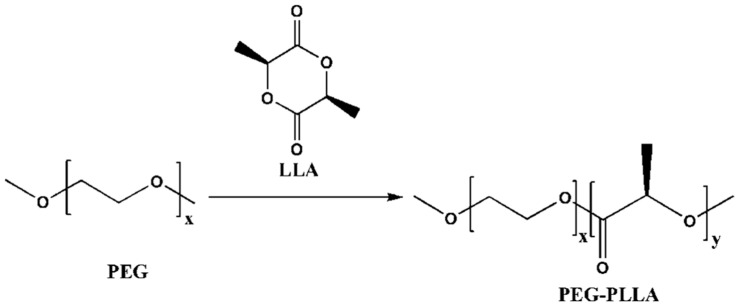
The synthesis route of PEG-PLLA diblock copolymer. PEG is used as an initiator, which undergoes a ring-opening reaction with lactide to obtain a diblock copolymer PEG-PLLA.

**Figure 3 pharmaceutics-15-02693-f003:**
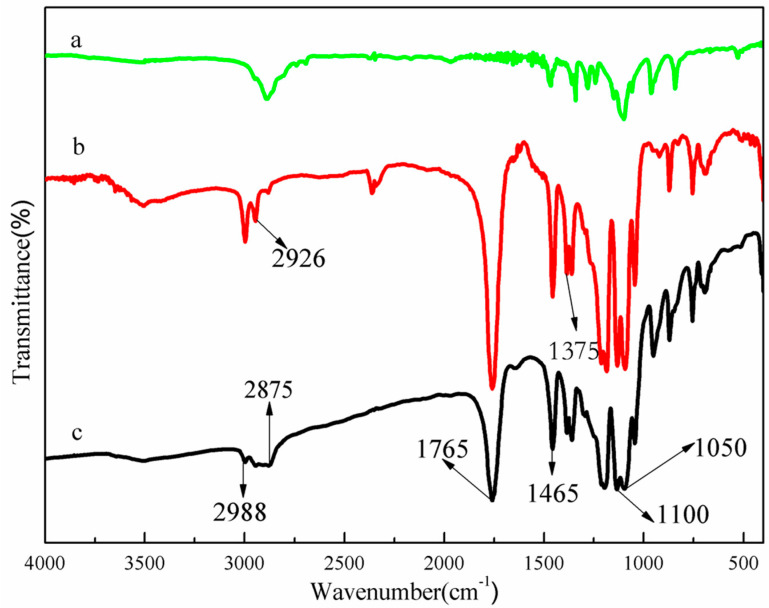
Fourier transform infrared spectra of PEG (a), PLLA (b), and PEG-PLLA (c).

**Figure 4 pharmaceutics-15-02693-f004:**
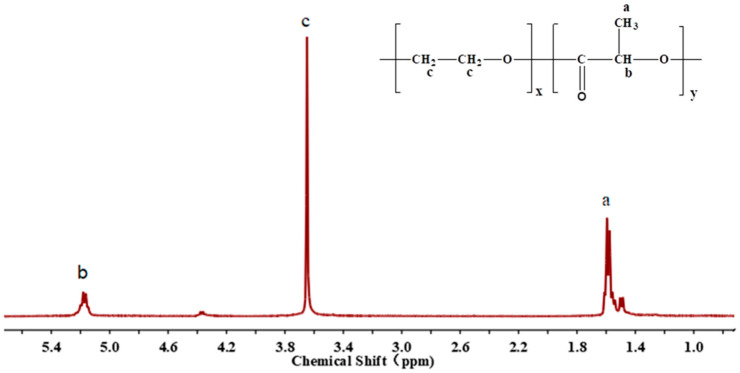
The proton nuclear magnetic resonance spectra (in CDCl_3_, ppm) of PEG-PLLA.

**Figure 5 pharmaceutics-15-02693-f005:**
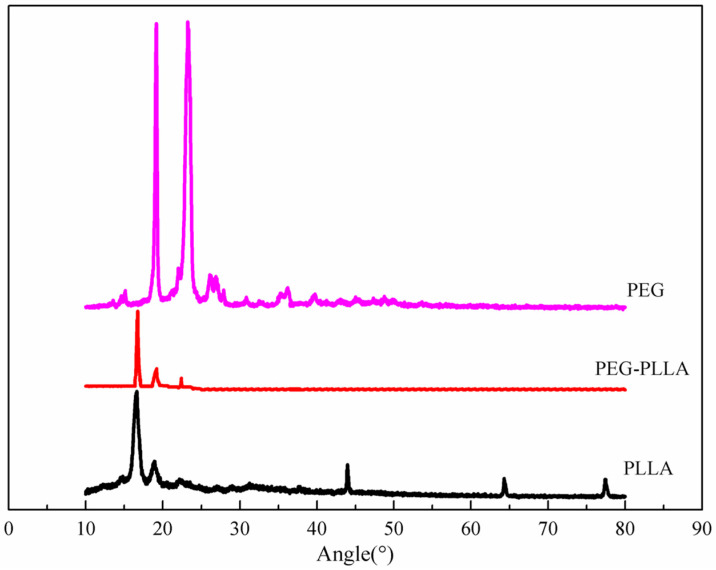
The XRD patterns of PLLA, PEG-PLLA, and PEG. The addition of PEG disrupts the crystal structure of PLLA, resulting in a decrease in the orderliness of the PLLA molecules.

**Figure 6 pharmaceutics-15-02693-f006:**
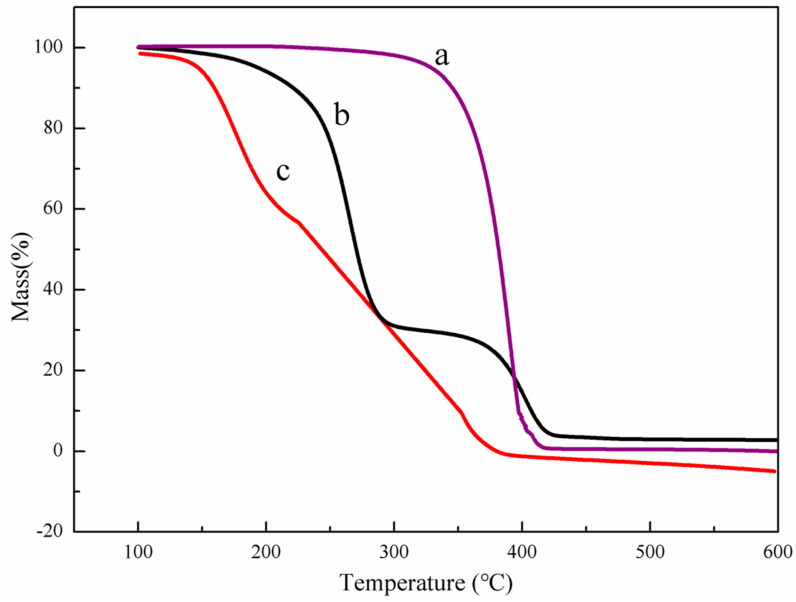
TGA curves of PEG (a), PEG-PLLA (b), and PLLA (c). The thermal weight loss of the sample at different temperatures can be observed within 100–600 °C.

**Figure 7 pharmaceutics-15-02693-f007:**
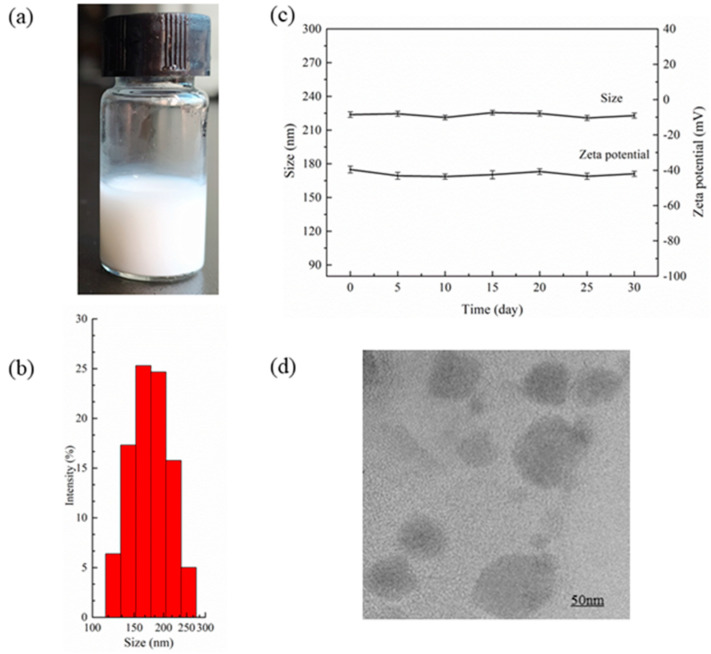
Characterization of the PEG-PLLA@SA·PFP nanoparticles. (**a**) Digital photo. (**b**) Particle size distribution. (**c**) The stability of the size and zeta potential of the nanoparticles in vitro. (**d**) TEM image of PEG-PLLA@SA·PFP nanoparticles.

**Figure 8 pharmaceutics-15-02693-f008:**
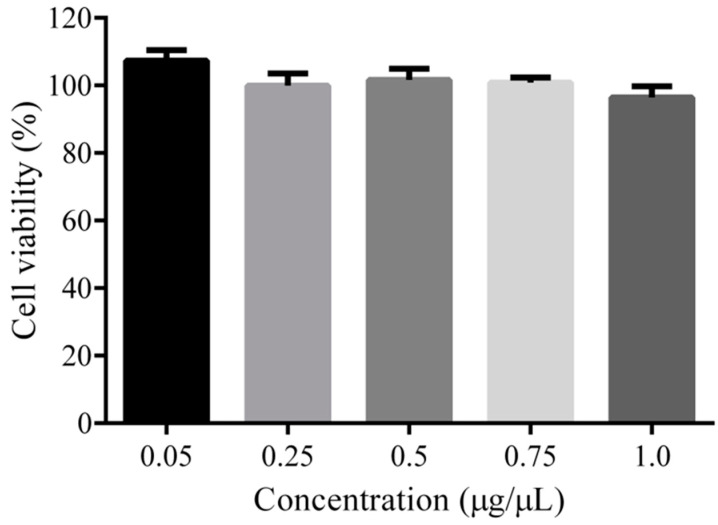
In vitro cytotoxicity with a CCK-8 assay. In vitro cell viability of C6 cells incubated with PEG-PLLA copolymers at different concentrations for 24 h and detected at a wavelength of 492 nm with a microplate reader. Data are reported as mean ± SD (n = 5).

**Figure 9 pharmaceutics-15-02693-f009:**
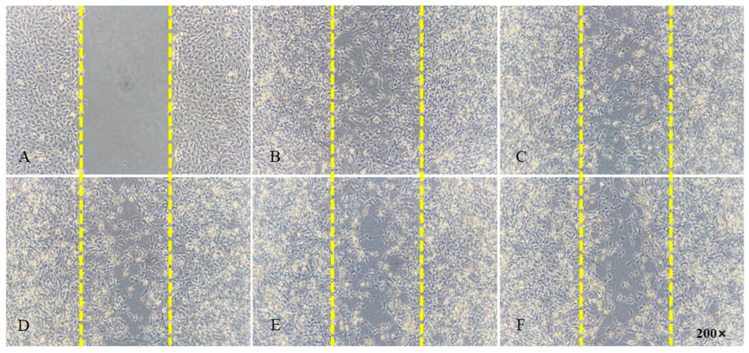
Effects of the PEG-PLLA copolymer on the migration of C6 cells. (**A**) Cultured for 0 h, (**B**) control group cultured for 24 h, (**C**) sample group (0.25 μg/μL) cultured for 24 h, (**D**) sample group (0.5 μg/μL) cultured for 24 h, (**E**) sample group (0.75 μg/μL) cultured for 24 h, (**F**) sample group (1.00 μg/μL) cultured for 24 h.

**Figure 10 pharmaceutics-15-02693-f010:**
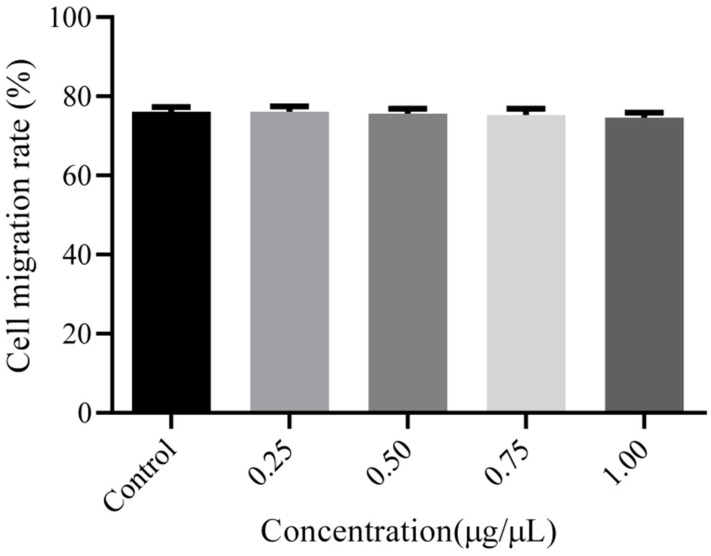
Cell migration rate of glioma cells with different PEG-PLLA copolymer concentrations after 24 h incubation. The results showed no significant differences (*p* > 0.05).

**Figure 11 pharmaceutics-15-02693-f011:**
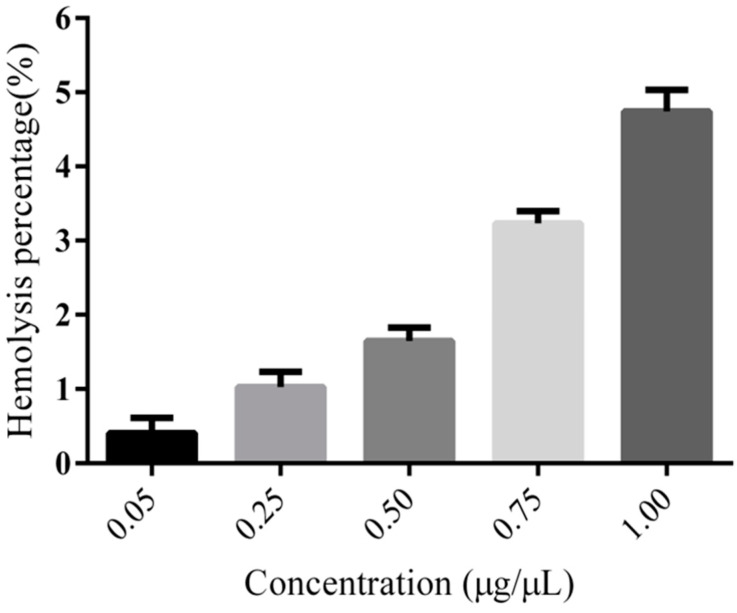
Hemolysis assay with different PEG-PLLA copolymer concentrations. Different concentrations of PEG-PLLA copolymer were mixed with red blood cells, and the destruction of the red blood cells was observed.

**Figure 12 pharmaceutics-15-02693-f012:**
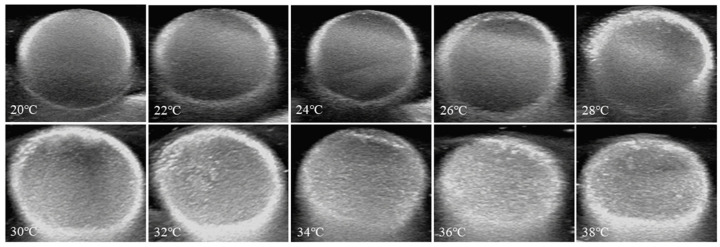
In vitro ultrasonic images of PEG-PLLA@SA·PFP nanoparticles in latex gloves, which were immersed in a thermostatic water bath. The temperature was increased from 20 °C to 38 °C at an interval of 2 °C.

**Figure 13 pharmaceutics-15-02693-f013:**
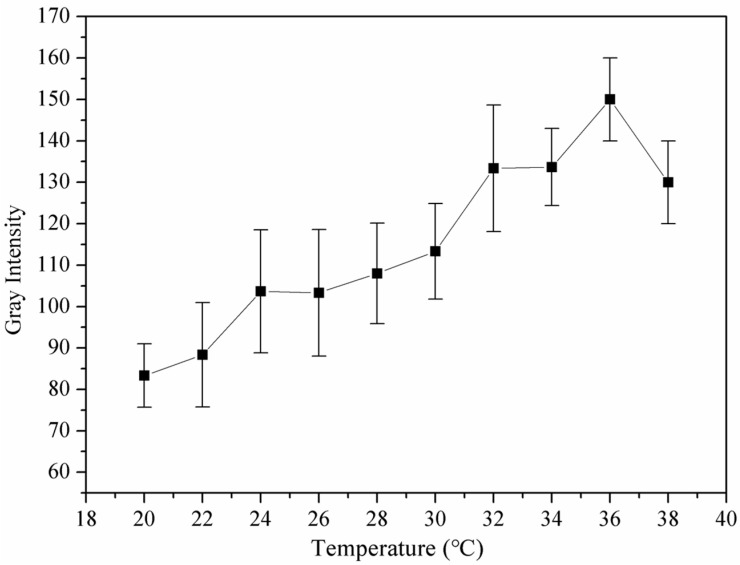
Representative in vitro ultrasonic images. Quantitative gray-scale for the ultrasonic intensity of PEG-PLLA@SA·PFP nanoparticles from 20 to 38 °C.

**Figure 14 pharmaceutics-15-02693-f014:**
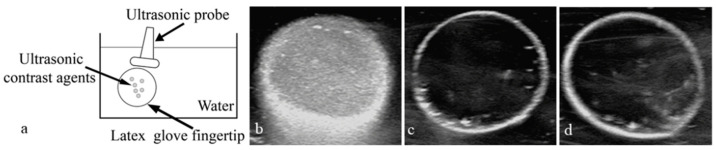
Experimental model of ultrasonic images in vitro. (**a**). Ultrasonic image of PEG-PLLA@SA·PFP nanoparticles (**b**), normal saline (**c**), and coupling agent (**d**) in the latex gloves mold under high-frequency diagnostic ultrasound at 37 °C.

**Figure 15 pharmaceutics-15-02693-f015:**
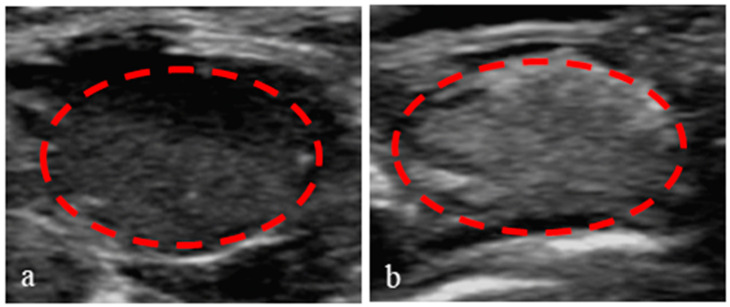
Ultrasound images of the subcutaneous tumor tissue in nude mice. Before (**a**) and after (**b**) the injection of PEG-PLLA@SA·PFP nanoparticles solution.

**Figure 16 pharmaceutics-15-02693-f016:**
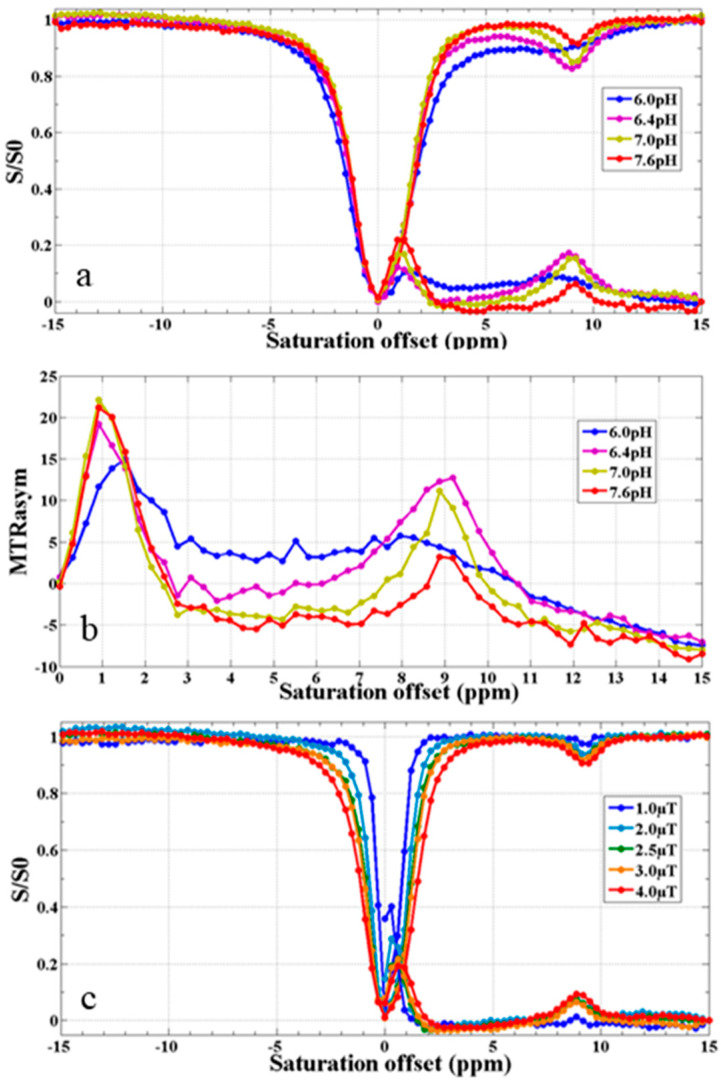
Z spectra (**a**) and MTR*_assy_* spectra (**b**) of PEG-PLLA@SA·PFP nanoparticles with different pHs (6.0, 6.4, 7.0, and 7.6). Z spectra (**c**) of nanoparticles with varying levels of the intensity of the saturated RF pulse (1.0, 2.0, 2.5, 3.0, and 4.0 μT).

**Figure 17 pharmaceutics-15-02693-f017:**
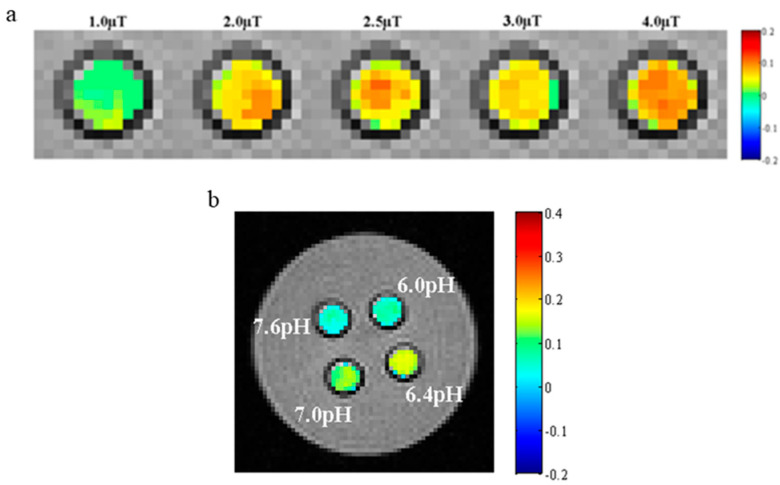
CEST images of PEG-PLLA@SA·PFP nanoparticles at different pHs (**a**) and intensities of saturated RF pulses (**b**) in vitro.

**Figure 18 pharmaceutics-15-02693-f018:**
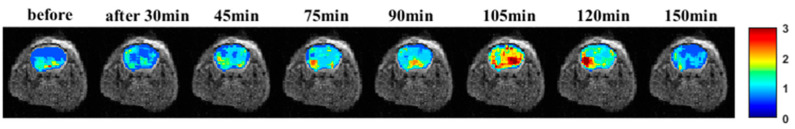
CEST images of brain gliomas in rats before and after injection of PEG-PLLA@SA·PFP nanoparticles.

## Data Availability

The data that support the findings of this study are available from the corresponding author, Yanlongjia (email to: yanlongjia@163.com) and Binhua, Luo (email to: luobinhua@hbust.edu.cn) upon reasonable request with the permission of the head of the department.

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
