# Peer review of "Novel Nanoprobe with Combined Ultrasonography/Chemical Exchange Saturation Transfer Magnetic Resonance Imaging for Precise Diagnosis of Tumors"

_pharmaceutics, 2023, doi:10.3390/pharmaceutics15122693_

Round 1
Reviewer 1 Report
Comments and Suggestions for Authors
The manuscript describes PEG-PLLA nanoparticle for MRI imaging. By loading drugs such as salicylic acid and perfluoropentane in the PEG-PLLA nanoparticles, the author tested the feasibility of dual mode imaging system including ultrasound imaging and CEST imageing.
The manuscript should be revised.
(1) In Figure 1, there is a schematic figure which describes the nanoparticle structure with the position of PFP and salicylic acids. Why did the author position the PFP in the center as a bulk shape, while the salicylic acid as dispersed particles around the PFP? Authors should explain rational or evidence about the schematic picture.
(2) About the analysis of the synthesized nanoparticle, authors should provide justifications of their synthesis. They only provide the data about PGA-PLLA. For the PFP-salicylic particle, author should provide several data including microscopic images like TEM images, or at least the solution (or powder) images for each step of the synthesis of PFP, Salicylic acid, and PEG-PLLA.
(3) About Figure 15, authors claimed that the nanoparticles were tested in nude mice. However, the information is not enough. (a) The transplanted tumor images should be provided with the whole picture of the mice. (b) A camera picture of the injection of tumor into the mice would be informative. (c) Author should provide the evidence of that the binding of the PFP-salicylic acid-PEG-PLLA nanoparticle is "tumor"-specific.
Comments on the Quality of English Language
English is fine.
Reviewer 2 Report
Comments and Suggestions for Authors
Dear colleagues,
In this manuscript, the authors demonstrate the possibility of analyzing medical images for cancer diagnosis and treatment monitoring which is crucial for modern medicine. The results are interesting. The figures reflect the results of the study. Despite the very good impression of the article, there are some questions which could improve the article in my opinion, partly:
Some terms require better interpretation, for example, “Ultrasound molecular imaging” and “Ultrasound” are not the same, so, they should be defined in the introduction.
That is unclear how was estimated conclusion about migration properties of the copolymer on C6 cells which was no significant difference in the cell migration. Quantitate data should be added for indicating that PEG-PLLA as an extracellular matrix did not affect cell migration. Figure 10 has only upper differences.
Only the upper margins of error are presented in all figures.
There is no discussion of the obtained results.
The conclusions should be changed with more digital results data.
The article requires serious stylistic correction.
In summary, I have been satisfied with the level of the article. I believe this manuscript will attract significant attention from the research community. In my personal opinion, the article is very valuable, a great prospect for further research, and, after corrections, can be recommended for publication.
The article requires serious stylistic correction.
Reviewer 3 Report
Comments and Suggestions for Authors
1. In Fig. 1, caption, change the 'nanoparticle' to the specific name of the nanohybrid developed.
2. All figure captions should be detailed enough to be self-understood independently
3. In Fig. 7 caption, instead of 'appearance', write 'digital photo'
4. In Fig. 9, put the scale bar
5. In Fig. 11, also put the digital photos of the hemolysis experiment
Comments on the Quality of English LanguageNo comments
Reviewer 4 Report
Comments and Suggestions for Authors
The authors described a new agent in the form of copolymer nanoparticles encapsulated with salicylic acid and perfluoropentane for ultrasound imaging and MRI. The characterization of the materials was thorough. The imaging protocols were clear. The conclusions were supported by sufficient experimental data. The manuscript was in general well written. So I think this work is ready for publication after a few minor modifications.
1. The temperature study in figure 13 should not be considered as optimal temperature at 37 C. This is because the numbers for 32-38 range are essentially the same. The statement should be weakened as “well suited for imaging at 37 C” or similar.
2. It will be helpful to readers not familiar with ultrasound imaging if the authors explain the role of perfluoropentane.
3. On the language, it will be valuable if the authors can polish a bit more. For example, Line 33, 66, and so on, they used “better” or “higher” etc. But what were those parameters compared to? So no need to use those words. But overall, nicely done.
Comments on the Quality of English LanguageGood overall.
Round 2
Reviewer 1 Report
Comments and Suggestions for Authors
All comments are reflected well in the revised manuscript.
Comments on the Quality of English LanguageEnglish is fine.
Reviewer 3 Report
Comments and Suggestions for Authors
Authors have addressed all my comments.